# Clinical Characterization of a 6-Year-Old Patient with Autism and Two Adjacent Duplications on 10q11.22q11.23. A Case Report

**DOI:** 10.3390/children8060518

**Published:** 2021-06-18

**Authors:** Giovanna Tritto, Ivana Ricca, Marco Turi, Andrea Gemma, Filippo Muratori, Gioacchino Scarano, Fortunato Lonardo

**Affiliations:** 1Fondazione Stella Maris Mediterraneo, 85100 Potenza, Italy; tritto.nina@gmail.com (G.T.); andrea.gemma@asmbasilicata.it (A.G.); 2Department of Developmental Neurocience, IRCCS Fondazione Stella Maris, 56128 Pisa, Italy; ricca.ivana@gmail.com (I.R.); filippo.muratori@fsm.unipi.it (F.M.); 3Medical Genetics Unit, A.O.R.N. San Pio, 82100 Benevento, Italy; gioac.scarano51@gmail.com (G.S.); fortunato.lonardo@gmail.com (F.L.)

**Keywords:** autistic spectrum disorders, chromosome 10, genomic microarray analysis

## Abstract

Autism is a neurodevelopmental disorder presenting in the first 3 years of life. Deficits occur in the core areas of social communication and interaction and restricted, repetitive patterns of behavior, interests or activities. The causes of autism are unknown, but clinical genetic studies show strong evidence in favor of the involvement of genetic factors in etiology. Molecular genetic studies report some associations with candidate genes, and candidate regions have emerged from several genome-wide linkage studies. Here, we report a clinical case of autism in a 6-year-old boy with double duplication on 10q11.22q11.23 with ASD (Autism Spectrum Disorder), intellectual disability, developmental delay, hypotonia, gross-motor skills deficit, overgrowth and mild dysmorphic features. In the literature, only five cases of ASD with 10q11.21q11.23 duplication are reported. This is the first extensive clinical description of an ASD subject with 10q11.22q11.23 duplication. Our findings suggest that 10q11.21q11.23 microduplication could represent a copy number variant that predisposes to autism.

## 1. Introduction

The clinical syndrome of autism was first described by Kanner in 1943. It is a neurodevelopmental disorder of childhood presenting with deficits in the three areas of communication, social interaction, and behavior.

The significant increase in awareness of the autism spectrum disorders and evolution of Diagnostic and Statistical Manual of Mental Disorders (DSM) criteria towards a core diagnosis covering a spectrum of disorders led to a significant increase of prevalence, estimated at around 62 cases per 10,000 people [1]. According to the current DSM-5 criteria, only two core features make up an autism spectrum disorder (ASD) diagnosis: (1) persistent impairments in social communication and social interaction across multiple contexts; and (2) restricted, repetitive patterns of behavior, interests, or activities [2].

Autism is now understood to be a disease of complex interaction between genetics and the environment, with heritability estimates ranging from 40 to 80% [3]. There is significant evidence from twin and family studies to suggest that genetic factors play a major role in the etiology [4,5].

In the beginning, the heritability of autism was estimated to be very low. However, a series of twin studies brought a radical change to these views, and findings consistently indicated increased concordance rates in monozygotic over dizygotic twins [4,6,7]. Based on these observations, genetic factors were initially estimated to explain over 90% of the liability to autism in the population. More recent twin studies reported a much lower estimate of autism heritability, ranging from 38% [8] to 83% [9]. Regardless of the exact magnitude of genetic effect, other observations have, in addition to the twin studies, provided further confirmation of a substantial role of genetic factors in ASD. Siblings of a child with ASD are at increased risk; the so-called recurrence rate is estimated to be between 10% and 19% [10], with a relative increase if the child with ASD is female [11]. At present, there is a growing list of genetic risk variants, for which the strong evidence of association with ASD is apparent in several independent studies. It is estimated that it is possible to identify a genetic contribution in approximately 20–30% of individuals with ASD [12] which, compared to the 2–3% of individuals with ASD in whom a genetic condition was identifiable up until 2005, represents a dramatic tenfold increase.

Epidemiological investigations have begun to elucidate which environmental factors might be contributing to risk, but there is still a lot to understand when it comes to how they interact with genetic predisposition to contribute to ASD etiology. Epigenetic influence, environmental factors and genetic-environment interactions are all supposed to be involved in the pathogenesis of ASD [13].

Linkage studies have found putative evidence for linkage on several regions, and association studies are ongoing [14,15,16]. More recently, the advent of genomic microarray analysis (GMA) and exome sequencing eased the identification of new autism-associated copy number variants (CNVs) and new candidate genes [17,18,19]. Several recurrent rearrangements have been added to the growing list of genomic disorders.

For example, regarding the 10q abnormalities, these include the 10q22.3q23.2 region comprising a complex set of low-copy repeats (LCRs) that may lead to various genomic alterations through non-allelic homologous recombination (NAHR) [20]. These deletions have been described to be associated with a wide range of cognitive and behavioral phenotypes, including learning difficulties, speech and language delay, ADHD (Attention Deficit Hyperactivity Disorder), dysmorphic features, cardiac defects, cerebellar anomalies, macrocephaly and autism [20]. Similarly, in a previous study [21], 4 cases with deletions involving the distal long arm of one chromosome 10 have been described, presenting heterogeneous clinical features.

Proximal 10q duplication is a rarer genetic syndrome, distinct from the more common distal 10q trisomy syndrome. In the past, many studies reported large proximal 10q duplications detected by karyotyping in patients with mild to moderate developmental delay, postnatal growth retardation, microcephaly, and dysmorphic features [22,23,24,25,26,27,28].

The use of GMA in clinical practice allows a more accurate definition of micro-deletions and duplication breakpoints, and a better genotype-phenotype characterization [29]. This is also true in autism spectrum disorders (ASD), where GMA is recommended as a first-tier analysis [30].

There are few reports of microduplications involving 10q11.22-q11.23 chromosomal bands detected by GMA. In 2014, Manolakos et al. [31] described the case of a 3-year-old boy with phenotypic abnormalities and duplication of the chromosomal region 10q11.21q11.22. A case of maternally inherited 10q11.23 duplication in an individual with autism was reported among novel potentially pathogenic CNVs found in the Childhood Autism Risks from Genetics and Environment (CHARGE) cohort [32]. Stankewicz et al. [33] reported 17 cases of microduplication 10q11.21q11.23 in patients with autism and development delay/intellectual disability. However, these reports do not provide precise clinical information to help define the phenotype associated with these CNVs. In 2018, Meguid et al. described the case of a child with duplication at 10q11.23-q23.2 detected by GMA presenting ASD and ID (Intellectual Disability) [34].

Here, we report a case of autism in a 6-year-old male patient associated with a double duplication on 10q11.22q11.23 and 10q11.23 detected by GMA.

This is the first extensive clinical and neuropsychological description of an ASD subject with 10q11.22q11.23 microduplication.

### Clinical History

Our 6-year-old patient is the third son of healthy non-consanguineous parents (44-year-old father and 43-year-old mother at the time of conception). He has two healthy brothers with normal psychomotor development history. An ectopic pregnancy occurred between the older brother and the child discussed here. Family history was negative except for a case of intellectual disability attributed to infantile meningitis (the brother of the paternal grandfather).

The parents reported an uneventful pregnancy and normal ultrasound examinations. The patient was born at the 35th week of gestational age by vaginal delivery. The Apgar scores were 9 (1st minute) and 9 (5th minute). Birth weight was 3450 g (>97° percentile), length 50 cm (90–97° percentile), and head circumference (HC) 35.5 cm (>97° percentile) [35]. At birth, he presented with transient hypoglycemia and unilateral cryptorchidism.

Poor breastfeeding latch, difficult weaning, and difficulties in accepting solid food were reported. He did not have sleep problems.

At 3 months of age, hypotonia, hyporeactivity, and poor visual involvement were noted. At the clinical evaluation, he showed some dysmorphic features: brachycephaly, blepharophimosis, flat nasal root, hypoplastic alae, bilateral brachytelephalangy of the first finger and single palmar crease.

Parents describe his first simple words at 12–13 months, then he showed very slow expressive language development. When he was 6 years old, he was able to occasionally say 5–6 single words to express his requests. Motor developmental delay was reported, requiring physical therapy: he was able to hold his head up at 12 months of age, trunk control was acquired at 24 months of age, and when he was 30 months old, he was able to walk alone.

At the time of the latest clinical and neurological evaluation (6 years of age), he presented with diffuse hypotonia, and gross motor skills deficits. Muscular trophism and strength, and pyramidal and cerebellar assessment, were normal. Dysmorphic features were: prominent forehead, bulbous nasal tip, anteverted nares, hypoplastic alae, single palmar crease, foetal pads, joint hyperlaxity, and mild dorsal hypertrichosis. Weight was 25 kg (75° percentile), height 133 cm (>97° percentile) [36] and HC 54 cm (97° percentile) [37]. Continence was not already reached.

Acoustic brain response (ABR), abdominal ultrasound, electrocardiogram, clinical cardiac examination, wake and sleep video-EEG, and neurometabolic screening were all normal. Ophthalmological examination reported a minimal optic disc pallor. Brain MRI (included the study of the optic nerve) was normal, except for a subtle and non-specific prominence of the upper edge of the pituitary gland.

## 2. Cognitive and Behavioral Assessment

After meeting the current DSM-5 criteria, autism spectrum disorder (ASD) severity was assessed when he was 6 years old with the semi-structured Autism Diagnostic Observation Schedule second edition (ADOS-2) evaluation [38]. Poor eye contact was detected. A few facial expressions were observed only for directing extreme emotional states. Interest in communication appeared severely limited. The child had no language, he mostly emitted vocalizations without communicative purposes, except for intense shouts when disappointed.

Use of another’s body as tool was mainly observed for reaching attractive objects. No pointing was observed; he could stretch his arm with an open hand without coordinated eye contact for asking. No gestures were observed except for refusing. He only interacted with others to achieve specific goals. He showed poor interest in toys, except for the ball. Repetitive patterns in play with the pop-up toy were present, and, during most of the observation, the child was repeatedly looking for the ball to kick. Recurring mannerisms were detected. The total ADOS-2 score was 21, which reflects a moderate level of severity of the symptoms associated with autism. The social affect (SA) and the restricted and repetitive behaviors (RRB) domains sub-scores were 17 and 4, respectively.

The assessment of the developmental profile at 4 years of age with the Griffith’s scale (GMDS) [39] failed because of the lack of interest in social interaction and activities and the presence of marked repetitiveness in playing. Cognitive assessment with the Leiter-R was unsuccessfully tried when he was 6 years old [40].

Standardized tests for assessing communication skills were not possible. Information about the child’s abilities in language, including vocabulary comprehension, production, gestures, and grammar, was obtained from the parents’ completed MacArthur Communicative Developmental Inventories [41]. Median equivalent age in Receptive Language, Expressive Language and Action and Gestures was between 14 and 16 months.

As mentioned above, due to the unfeseability of developmental/cognitive assessment, the PEP-3 was performed [42]. The PEP-3 evaluates the range of behaviors and abilities of children with ASD. Table 1 reports the patient’s scores in each scale on the PEP-3; note that the PEP-3 does not provide equivalent developmental age for AE, SR, CMB or CVB subscales.

Vineland-II Survey Interview Form (VABS-II) was compiled by a psychologist who interviewed the patient’s mother [43] for measuring adaptive functioning. The IQ score of the Composite Scale was equal to 30 (greater than 2 standard deviations below the norm). In individual scales, the child obtained an IQ of: 32 (<2SD) in Communication, 54 (<2SD) in Ability in everyday life, 53 (<2SD) in Socialization, and 32 (<2SD) in Motor Activity. Further details about the results obtained are reported in the Table 2.

## 3. Genetic Studies

Screening for CGG Repeat Expansion in the *FMR1* gene excluded Fragile X syndrome. Karyotype analysis showed that a 10q11.2 duplication [46, XY, dup(10) (q11.2q11.2)]. GMA was performed on DNA isolated from peripheral blood of the patient and both parents to define the breakpoints of the duplication and the familial segregation and revealed two maternal CNVs involving the 10q11.2 chromosomal region: a duplication of about 1.9 Mb involving the 10q11.22-q11.23 chromosomal bands (breakpoints chr10:49201549-51128136) and a 430 Kb duplication in the 10q11.23 region (breakpoints chr10:52002855-52432961) (Human Genome build 37, hg 19). The first larger and more proximal duplication is not reported in the healthy population (Database of Genomic Variants, DVG, http://dgv.tcag.ca/dgv/app/home?ref=GRCh37/hg19) (accessed on 24 May 2021) and spans about 20 genes. The second more distal CNV involves only 2 genes. None of the genes encompassed by these two CNVs is associated with neurodevelopmental disorders. However, the *WDFY4* gene is reported as a “strong candidate” ASD gene in the SFARI Gene database (https://gene.sfari.org/) (accessed on 24 May 2021).

GMA results are depicted in Figure 1.

## 4. Discussion

In recent years, the findings of genetic studies in ASD increased the interest in the role of genetic factors in the etiology of this disorder. A genetic cause can be identified in 20% to 25% of children with autism [44]. Cytogenetically visible chromosomal abnormalities (~5%), copy number variants (CNVs) (10–20%) and single gene disorders in which neurologic findings are associated with ASD (~5%) are among the known genetic causes of ASD [44]. The most extensive use of GMA in clinical practice has increased descriptions of many and different phenotypes in individuals with ASD in literature.

Identifying the full range of ASD phenotypes affected by genetic variants could help to: (a) organize active surveillance and early intervention for individuals that present genetic risk; (b) predict cognitive and behavioral profile that can direct specific treatment; (c) reduce families’ stress levels by knowing the cause of the ASD in their children [45].

Finally, the increasing knowledge of genetic variants and associated phenotypes may suggest or increase underlying biological mechanisms.

The present study describes the case of a 6-year-old patient with ASD, intellectual disability, developmental delay history, hypotonia, gross motor skills deficits, overgrowth and mild dysmorphic features.

The first case of proximal duplication of 10q was described by Vogel in 1978. Since then, to our knowledge, only a few additional cases with 10q11q22 duplications have been reported [22,23,24,27,46]. These authors defined the “proximal trisomy 10q syndrome”, consisting of mild to moderate developmental delay, postnatal growth retardation, microcephaly, and dysmorphic features. Some of the clinical features described in the previously published cases as developmental delay and some dysmorphic features (deep-set, small eyes; blepharophimosis; hypermobile joints) are shared with the present case. However, these cases were studied with standard karyotype, which did not allow a precise definition of the duplication’s breakpoints and therefore the characterization of the phenotype-genotype relation. In our case, the GMA analysis revealed the presence of two duplications involving the 10q11.2 chromosomal region and both inherited from the mother.

To our knowledge, in the literature, less than 20 cases of 10q proximal duplication studied with GMA were reported. In 2014, Manolakos et al. [31] described a 3-year-old boy with a 10q11.21q11.22 duplication, which showed hypotonia, developmental delay, and mild dysmorphic features (enlarged head circumference, cryptorchidism, and single palmar creases of hands). In their study, the authors did not report autism. Conversely, they described the patient without social deficit. Manolakos et al. provided the first clinical phenotype description of a child with a 10q11.21q11.22 duplication detected by GMA. Instead, our case report is the first complete clinical phenotype description of a child with a similar duplication on 10q that present with autism assessed with gold standard tools.

As mentioned above in literature, there are studies [32,33,34] that show microduplications on 10q11.21q11.23 region associated with autism, however the authors did not extensively describe phenotypes, but reported some clinical features. For example, Stankiewicz et al. [33] reported the molecular characterization of 17 patients with microduplication in 10q11.21q11.23. Similar to our patient, most of them (11/17) had DD/ID and four had autism. However, these authors did not find statistically significant differences in duplication prevalence between the affected and control populations. Interestingly, 6 of 9 of the known duplications were maternally inherited as in our case. The authors did not report any extensive clinical description of the four cases with autism and they did not report assessment tools used for ASD diagnosis. These authors showed that deletions and duplications involving the 10q11.21q11.23 chromosomal regions are recurrent genomic events mediated by low-copy repeats.

An additional case of autism and a maternally inherited 10q11.23 duplication was described by Girirajan et al. [32], but similarly to Stankiewicz et al. [33], they did not report other clinical details.

In 2018, Meguid et al. [34] described the case of a 4-year-old boy with a heterozygous duplication at 10q11.23-q23.2 detected by GMA. The case described had a history of developmental delay, hypotonia and macrocephaly. Authors reported a diagnosis of secondary autism as defined in DSM-V, using gold standard assessment tools (ADI-R, CARS and ADOS); intellectual disability was detected, evaluated by using the Stanford–Binet scale. They reported a history of epileptic seizures, generalized epileptic foci on EEG and MRI abnormalities. In summary, Meguid et al. recently provided the first wide description of a child with a duplication at 10q11.23-q23.2 presenting autism and intellectual disability.

Differently from our case, the microduplication (10q11.23-q23.2) described by Meguid et al. [34] is larger and does not overlap with the microduplication (10q11.21q11.23) that we found in our case.

These three studies suggested the possible association between autism and 10q11.21q11.23 duplications. Unfortunately, comparing detailed clinical and neuropsychological findings of the previous two studies’ [32,33] patients with our case is not possible because of them not being available. The case described by Meguid et al. [34] showed ASD and ID, but it is not comparable with our case report, because the microduplications involved are adjacent but do not overlap.

## 5. Conclusions

In conclusion, this is the first extensive clinical description of an ASD subject with 10q11.22q11.23 duplication. Our paper contributes to better defining the phenotype associated with 10q11.2 duplications and to increase knowledge about the possible association between these CNVs and autism. In the duplicated regions, there are several genes, so the genotype-phenotype characterization is difficult. Only *WDFY4* is a possible autism-candidate gene reported in the SFARI Gene database. The microduplication detected in our child is maternally inherited, and other factors could have played a role in autism pathogenesis; among them, advanced parental age should be taken into consideration [47].

More clinical descriptions of individuals with microduplication 10q11.21q11.23 are needed to clarify the pathogenicity and penetrance of these microduplications. To explain the phenotypic variability and incomplete penetrance of these CNVs, we cannot exclude the presence of secondary mutational hits at other loci, which is a well-known mechanism of phenotypic variability associated with pathogenic CNVs [48]. Moreover, GMA cannot detect point mutations, which can only be searched for with sequencing techniques.

## Figures and Tables

**Figure 1 children-08-00518-f001:**
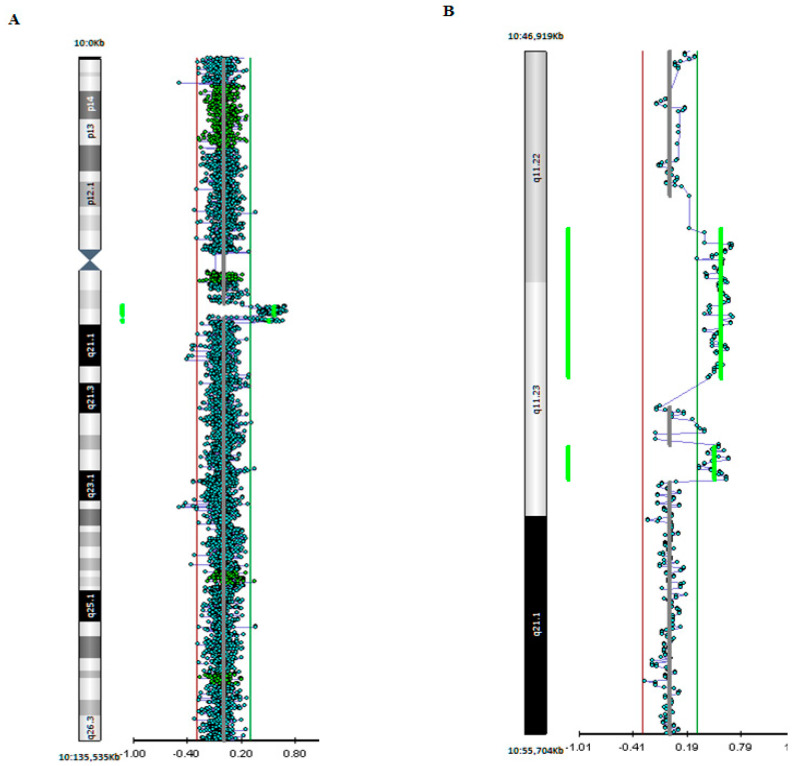
Array-CGH profile showing the 10q11 duplications detected in our patient. (**A**) View of the whole chromosome 10 and of the chromosomal region involved in the duplication. (**B**) Enlarged view showing the two adjacent duplications on 10q11.22q11.23.

**Table 1 children-08-00518-t001:** Patient’s scores on the PEP-3.

Scales	Percentile	Developmental Age (Months)
Cognitive verbal/pre-verbal (CVP)	7	27
Expressive language (EL)	2	<12
Receptive language (RL)	7	18
Fine motor (FM)	9	29
Gross motor (GM)	13	29
Visual-motor imitation (VMI)	9	24
Affective expression (AE)	15	
Social reciprocity (SR),	41	
Characteristic motor behaviour (CMB)	9	
Characteristic verbal behaviour (CVB)	11	

**Table 2 children-08-00518-t002:** Patient’s scores on the VABS-II.

	Subscale	Equivalent Age (Months)
Communication	Receptive	13
	Expressive	18
	Written	43
Daily living skills	Personal	27
	Domestic	33
	Community	18
Socialization	Interpersonal	17
	Play and leisure time	24
	Coping skills	13
Motor skills	Gross	42
	Fine	31

## Data Availability

The data presented in this study are available upon request from the corresponding author. The data are not publicly available due to privacy restrictions.

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
