# Peer review of "Clinical Characterization of a 6-Year-Old Patient with Autism and Two Adjacent Duplications on 10q11.22q11.23. A Case Report"

_children, 2021, doi:10.3390/children8060518_

Round 1

Reviewer 1 Report

Re: children-1210380

The authors clinically characterize a case of a specific set of duplications with potential relevance to autism. It is of interest at the case report level. There are just some issues with its communication.

Introduction- Should mention the DSM V criteria in the first paragraph to follow on the Kanner description, and discuss the complexity of the etiology of autism- the way it is written, it looks like we just counting up genes with larger studies- but it is a bit more complicated than that.  Change ‘3-years-old’ to ‘3-year-old’, and change ‘maternal inherited’ to ‘maternally inherited’.

Clinical history- change ‘years-old’ to ‘year-old’ as with the Introduction.  Was the ectopic pregnancy between the older brothers and the current case?  Just rephrase for clarity.  Change ‘presented transient’ to ‘presented with transient’, rephrase ‘hypovalid suction’, insert ‘and a’ before ‘single palmar crease’, something is wrong with the sentence ending ‘for asking at times’- please clarify, and insert ‘with’ before ‘diffuse hypotonia’. 

Section 2- Might rephrase how the diagnosis was made.   Maybe ‘After meeting the current DSM-5 criteria’ instead.  Please clarify ‘Few facial expressions were seen for directing extreme emotions’.  On multiple occasions change ‘kid’ to ‘child’.  Clarify – are the emitted vocalizations non-language vocalizations?  Clarify ‘Use of the other’- do the authors mean ‘Use of another’s body as a tool’?  Change ‘Pointing is’ to ‘Pointing was’.  Do the authors mean ‘or kicking it’ rather than ‘for kicking it’?  Insert ‘the’ before ‘Leiter-R’, insert ‘the’ before ‘child’s abilities in language’, change ‘by the parents competed’ to ‘from the parents who competed the’.  Change ‘unfeasible’ to ‘unfeasibility’, change ‘has been performed’ to ‘was performed’, clarify ‘fan of behaviors’, insert ‘the’ before ‘PEP-3’.  For Tables 1 and 2- change ‘in’ to ‘on the’.  Clarify ‘was equal to 30 (less than 2 standard deviations)’- do the authors mean ‘greater than 2 standard deviations below the norm’ in the parentheses?  Clarify ‘IQ of deviation of’- would just ‘IQ’ suffice here?

Discussion- change ‘condition’ to conditions’, change ‘profile’ to ‘profiles’, insert ‘and’ before ‘let many families’, and rephrase ‘Not at least’ in paragraph 1.  Insert a space before 1978, and change ‘duplications breakpoints’ to ‘duplications’ breakpoints’, as it is a possessive.  Since the authors were pointing out what is missing in the literature, do they mean ‘and sociability was not described’ instead of ‘sociability was described’?  Also, was there anything special about this region that makes it susceptible to multiple breakpoints?  Finally, it seems worth mentioning the parents’ relatively advanced age as a risk factor somewhere.

Conclusions- at the very end, it seems a PMID was left in instead of the reference- please fix.

Author Response

Dear Editor,

We are submitting the revised version of the paper. In revising the text, we took into account all the comments raised by the Reviewers. We thank for the comments that helped us to strengthen the manuscript and hope that is it now suitable for publication in Children.

Reviewer 1

The authors clinically characterize a case of a specific set of duplications with potential relevance to autism. It is of interest at the case report level. There are just some issues with its communication.

We thank the reviewer to appreciate or work.

Introduction- Should mention the DSM V criteria in the first paragraph to follow on the Kanner description, and discuss the complexity of the etiology of autism- the way it is written, it looks like we just counting up genes with larger studies- but it is a bit more complicated than that.  Change ‘3-years-old’ to ‘3-year-old’, and change ‘maternal inherited’ to ‘maternally inherited’.

Now we have  mentioned the DSM-V criteria and discuss the complexity of ASD and its etiology in the introduction.

“The significant increase in awareness of the autistic conditions and evolvement of Diagnostic and Statistical Manual of Mental Disorders (DSM) criteria towards a core di-agnosis covering a spectrum of disorders led to a significante increase of prevalence, es-timated around 62 cases per 10,000 people. According to the current DSM-5 criteria, only two core features make up an autism spectrum disorder (ASD) diagnosis: (1) per-sistent impairments in social communication and social interaction across multiple con-texts; and (2) restricted, repetitive patterns of behavior, interests, or activities. Autism is now understood to be a disease of complex interaction between genet-ics and the environment, with heritability estimates ranging from 40 to 80%. At the beginning, the heritability of autism was estimated to be very low. How-ever, a series of twin studies brought a radical change to this views findings consistently indicated increased concordance rates in monozygotic over dizygotic twins. Based on these observations, genetic factors were initially estimated to explain over 90% of the liability to autism in the population.  More recent twin studies reported a much lower estimate of autism heritability ranging from 38%8 to 83% . Regardless of the exact magnitude of genetic effect, other observations have, in addition to the twin studies, provided further confirmation of a substantial role of genetic factors in ASD. Siblings of a child with ASD are at increased risk; the so-called recurrence rate is estimated to be between 10% and ~19% with a relative increase if the child with ASD is female. At present, there is a growing list of genetic risk variants for which strong evidence of association with ASD is apparent in several independent studies. It is estimated that it is possible to identify a genetic contribution in approximately 20–30% of individuals with ASD which, compared to the ~2–3% of autistic individuals in whom a genetic condition was identifiable up until 2005, represents a dramatic tenfold increase. Epidemiological investigations have begun to elucidate which environmental factors might be contributing to risk, but there is a lot left to understand about how they interact with genetic predisposition to contribute to ASD etiology. Epigenetic influence, environmental factors and genetic-environment interactions are all supposed to be involved in the pathogenesis of ASD.”

We also changed ‘3-years-old’ to ‘3-year-old’ and change ‘maternal inherited’ to ‘maternally inherited’.

Clinical history- change ‘years-old’ to ‘year-old’ as with the Introduction.  Was the ectopic pregnancy between the older brothers and the current case?  Just rephrase for clarity.  Change ‘presented transient’ to ‘presented with transient’, rephrase ‘hypovalid suction’, insert ‘and a’ before ‘single palmar crease’, something is wrong with the sentence ending ‘for asking at times’- please clarify, and insert ‘with’ before ‘diffuse hypotonia’. 

Now we have corrected according with reviewer’s suggestions.

Section 2- Might rephrase how the diagnosis was made.   Maybe ‘After meeting the current DSM-5 criteria’ instead.  Please clarify ‘Few facial expressions were seen for directing extreme emotions’.  On multiple occasions change ‘kid’ to ‘child’.  Clarify – are the emitted vocalizations non-language vocalizations?  Clarify ‘Use of the other’- do the authors mean ‘Use of another’s body as a tool’?  Change ‘Pointing is’ to ‘Pointing was’.  Do the authors mean ‘or kicking it’ rather than ‘for kicking it’?  Insert ‘the’ before ‘Leiter-R’, insert ‘the’ before ‘child’s abilities in language’, change ‘by the parents competed’ to ‘from the parents who competed the’.  Change ‘unfeasible’ to ‘unfeasibility’, change ‘has been performed’ to ‘was performed’, clarify ‘fan of behaviors’, insert ‘the’ before ‘PEP-3’.  For Tables 1 and 2- change ‘in’ to ‘on the’.  Clarify ‘was equal to 30 (less than 2 standard deviations)’- do the authors mean ‘greater than 2 standard deviations below the norm’ in the parentheses?  Clarify ‘IQ of deviation of’- would just ‘IQ’ suffice here?

 Now we have corrected according with reviewer’s suggestions.

Discussion- change ‘condition’ to conditions’, change ‘profile’ to ‘profiles’, insert ‘and’ before ‘let many families’, and rephrase ‘Not at least’ in paragraph 1.  Insert a space before 1978, and change ‘duplications breakpoints’ to ‘duplications’ breakpoints’, as it is a possessive.  Since the authors were pointing out what is missing in the literature, do they mean ‘and sociability was not described’ instead of ‘sociability was described’? 

 Now we have corrected according with reviewer’s suggestions.

Also, was there anything special about this region that makes it susceptible to multiple breakpoints? 

Now we discuss this point in the text:

”These authors showed that deletions and duplications involving the 10q11.21q11.23 chromosomal regions are recurrent genomic events mediated by low-copy repeats.”

Finally, it seems worth mentioning the parents’ relatively advanced age as a risk factor somewhere.

Now we discuss this point in the text:

“The microduplication detected in our child is maternally inherited and other factors could have had a role in autism pathogenesis; among them the advanced parents age should be taken in consideration. “

Conclusions- at the very end, it seems a PMID was left in instead of the reference- please fix.

Thank the reviewer to find out this mistake, now we fixed it.

Reviewer 2 Report

The way that autism spectrum disorders is defined and described has culturally altered over the years so whilst it is interesting to see the reference to Kanner, a more up to date description would be better using DSM5 criteria as this is based on the latest evidence. In particular I would drop the word 'deficits'. There is growing evidence that neurodiversity exists based on evolutionary pressures and that there are strengths and challenges living in modern society with an ASD. I would therefore write about 'differences' and describe what these are. 

I am not keen on  the term 'hypovalid suction'. It is not a widely used term and think that we should be using terms that families reading information about their child would not find offensive. I think there are better ways. Perhaps change kid for child  for example. Similarly I would rephrase 'the kid mostly emitted vocalizations'.

Describe what you mean by 'jargon was observed'. 6 year old children often use interesting or exploratory words and phrases.

Is there a reason that some developmental age scores are not in table 1 (explain this in the  text if so).

The first paragraph of the discussion is difficult to  follow. Please can you check

In the discussion you need to discuss the differences between Stankiewicz et al 2012 and your findings more.

What does your study add?

I expecting to see references such as Van  Bon 2011;  and Yatsenko et al, 2009 on 10q. The discussion also needs to have some references on phenotypic variation from 10q duplications   (e.g. Yu et al 2019). Very important.

Please confirm that you have fully informed consent from both parents to publish details about this child.

Author Response

Dear Editor,

We are submitting the revised version of the paper. In revising the text, we took into account all the comments raised by the Reviewers. We thank for the comments that helped us to strengthen the manuscript and hope that is it now suitable for publication in Children.

Reviewer 2

The way that autism spectrum disorders is defined and described has culturally altered over the years so whilst it is interesting to see the reference to Kanner, a more up to date description would be better using DSM5 criteria as this is based on the latest evidence. In particular I would drop the word 'deficits'. There is growing evidence that neurodiversity exists based on evolutionary pressures and that there are strengths and challenges living in modern society with an ASD. I would therefore write about 'differences' and describe what these are. 

Now we have discussed in the introduction DSM criteria and corrected the term “deficits”. We added some details about ASD criteria in the introduction. See reply to referee 1.

I am not keen on  the term 'hypovalid suction'. It is not a widely used term and think that we should be using terms that families reading information about their child would not find offensive. I think there are better ways. Perhaps change kid for child  for example. Similarly I would rephrase 'the kid mostly emitted vocalizations'.,

Now we have rephrased both of them for more clarity.

“Poor breastfeeding latch”;

“Interest in communication appeared severely limited. The kid had no language, he mostly emitted vocalizations without communicative purpose except for intense shouts when disappointed.”

Describe what you mean by 'jargon was observed'. 6 year old children often use interesting or exploratory words and phrases.

Now we have rephrased in :

“he mostly emitted vocalizations without communicative purpose except for intense shouts when disappointed”

Is there a reason that some developmental age scores are not in table 1 (explain this in the  text if so).

Now it is explained in the text.

The first paragraph of the discussion is difficult to  follow. Please can you check

We tried to synthesize and reorganize the content.

“In the latest years the findings of genetic studies in ASD increased the interest in the role of genetic factors in the eziology of this disorder. A genetic cause can be identified in 20% to 25% of children with autism. Cytogenetically visible chromosomal abnormali-ties (~5%), copy number variants (CNVs) (10-20%) and single gene disorders in which neurologic findings are associated with ASD (~5%) are among the known genetic causes of ASD. The most extensive use of GMA in clinical practice has increased descriptions of many different phenotyopes in individuals with ASD in literature. Identifying the full range of ASD phenotypes affected by genetic variants could help to: a) organize active surveillance and early intervention for individuals that pre-sent genetic risk; b) predict cognitive and behavioral profile that can direct specific treatment; c) reduce families stress levels by knowing the cause of the ASD in their children.”

In the discussion you need to discuss the differences between Stankiewicz et al 2012 and your findings more.

Now we have expanded the discussion about differences between Stankiewicz et al 2012 and our findings.

“As mentioned above in literature there are studies that show microduplications on 10q11.21q11.23 region associated with autism, however the authors did not extensively described phenotypes but reported some clinical features. For example Stankie-wicz et al. reported the molecular characterization of 17 patients with microduplication in 10q11.21q11.23. Similar to our patient, most of them (11/17) had DD/ID and four had autism. However, these authors did not find statistically significant differences in dupli-cation prevalence between the affected and the control populations. Interestingly, 6 of 9  of the known duplications were maternally-inherited as in our case. Authors did not report extensive clinical description of the four cases with autism and they did not re-port assessment tools used for ASD diagnosis.  These authors showed that deletions and duplications involving the 10q11.21q11.23 chromosomal regions are recurrent genomic events mediated by low-copy repeats.”

What does your study add?

Now we reported clearer the main finding of our work:

It is expressed in the initial part of the conclusion and in the discussion.

"It is the first extensive clinical description of the phenotype of a child with microduplication 10q11.21q11.23 with ASD.”

I expecting to see references such as Van  Bon 2011;  and Yatsenko et al, 2009 on 10q. The discussion also needs to have some references on phenotypic variation from 10q duplications   (e.g. Yu et al 2019). Very important.

Now we have expanded the introduction and added references suggested. However we have problem to find Yu’s (2019) reference. Can the reviewer indicate the whole title of the paper, in order to include it in the discussion?

“Several recurrent rearrangements have been added to the growing list of genomic disorders. For example regarding the 10q abnormalities, these includes 10q22.3q23.2 region comprising a complex set of low-copy repeats (LCRs) that may lead to various genomic alteration through non-allelic homologous recombination (NAHR). These deletions have been described to be associated with a wide range of cogni-tive and behavioral phenotypes, including learning difficulties, speech and language delay, ADHD, dysmorphic features, cardiac defects, cerebellar anomalies, macroceph-aly and autism.  Similarly in previously study have been described 4 cases with deletions involving the distal long arm of one chromosome 10, presenting heterogeneous clinical features.”

Please confirm that you have fully informed consent from both parents to publish details about this child.

We certainly have it. We have sent it to the Editor.

Round 2

Reviewer 1 Report

Re: children-1210380

The authors clinically characterize a case of a specific set of duplications with potential relevance to autism. It is of interest at the case report level. There are just some issues with its communication, many of the issues coming in the new text.

Introduction- in the new text: Change ‘significante’ to ‘significant’.  Fix the odd text ‘However[GT1}’.  Change ‘these includes’ to ‘these include the’.  Fix grammar for ‘Similarly in a previous study have been described…’  Change ‘kid’ to ‘child’.

Clinical history- a few of my comments were missed:  something is still wrong with the sentence ending ‘for asking at times’- please clarify, and insert ‘with’ before ‘diffuse hypotonia’. 

Section 2- Might rephrase how the diagnosis was made.   Maybe ‘After meeting the current DSM-5 criteria’ instead of ‘According to…’ as the ADOS is confirming DSM-5.  Change ‘kid’ to ‘child’- multiple occasions.  ‘Kid’ is not appropriate for a professional journal.  Change ‘in each scale in PEP-3’ to ‘in each scale on the PEP-3’. 

Discussion- change ‘eziology’ to ‘etiology’.   Change ‘families stress levels’ to ‘families’ stress levels’.  Rephrase ‘Not at least’ in paragraph 3- maybe ‘at the very least’ would be better.  PLEASE change ‘kid’ to ‘child’ in this section too- multiple occasions.  Change ‘present autism’ to ‘presents with autism’.    Change ‘at EEG, MRI’ to ‘on EEG, and MRI’  For ‘is more wide’ do the authors mean ‘is larger’?  Change ‘previous two studies patients’ to ‘previous two studies’ patients’.

Author Response

The authors clinically characterize a case of a specific set of duplications with potential relevance to autism. It is of interest at the case report level. There are just some issues with its communication, many of the issues coming in the new text.

Introduction- in the new text: Change ‘significante’ to ‘significant’.  Fix the odd text ‘However[GT1}’.  Change ‘these includes’ to ‘these include the’.  Fix grammar for ‘Similarly in a previous study have been described…’  Change ‘kid’ to ‘child’.

Thank you, now we have corrected according with reviewer’s suggestions. We also changed ‘kid’ to ‘child’ in the text.

Clinical history- a few of my comments were missed:  something is still wrong with the sentence ending ‘for asking at times’- please clarify, and insert ‘with’ before ‘diffuse hypotonia’. 

Now we changed “for asking at times” to “to occasionally say 5-6 single words to express his requests”. We also inserted “with” before ‘diffuse hypotonia’. 

Section 2- Might rephrase how the diagnosis was made.   Maybe ‘After meeting the current DSM-5 criteria’ instead of ‘According to…’ as the ADOS is confirming DSM-5.  Change ‘kid’ to ‘child’- multiple occasions.  ‘Kid’ is not appropriate for a professional journal.  Change ‘in each scale in PEP-3’ to ‘in each scale on the PEP-3’. 

 Now we changed ‘According to…’ with ‘After meeting the current DSM-5 criteria’. We also changed ‘kid’ to ‘child’ in the text.

Discussion- change ‘eziology’ to ‘etiology’.   Change ‘families stress levels’ to ‘families’ stress levels’.  Rephrase ‘Not at least’ in paragraph 3- maybe ‘at the very least’ would be better.  PLEASE change ‘kid’ to ‘child’ in this section too- multiple occasions.  Change ‘present autism’ to ‘presents with autism’.    Change ‘at EEG, MRI’ to ‘on EEG, and MRI’  For ‘is more wide’ do the authors mean ‘is larger’?  Change ‘previous two studies patients’ to ‘previous two studies’ patients’.

Now we have corrected according with reviewer’s suggestions.

Reviewer 2 Report

Thank you for the additions.

These need proof reading more carefully in general. Please also include correction of the following errors:

Intro para 2 'autistic conditions' should be 'autism spectrum disorders'

Intro para 2 'evolvement' should be 'evolution'

Intro para 4 '...a radical change to this views findings consistently indicated..' (this sentence needs to be rewritten)

Intro para 4 should 'autistic individuals' read 'individuals with ASD' (switch in terminology in confusing)

Intro para 7 'similarly in previous study have been described ..' should be '..similarly in a previous study 4 cases with deletions have been described' (please check carefully)

Intro para 10 please remove the word 'kid' and change to 'child' (do this throughout the paper)

Section 1.1: Change to: 'An ectopic pregnancy occurred between the older brother and the child discussed here'

Discussion second sentence: please provide a reference.

Discussion (again change 'kid' to 'child')

Discussion para 6 Please provide Manolakos reference

Discussion para 7 Please check English and also second last paragraph in Discussion

Author Response

These need proof reading more carefully in general. Please also include correction of the following errors:

Intro para 2 'autistic conditions' should be 'autism spectrum disorders'

Thank you, now we have corrected according with reviewer’s suggestions

Intro para 2 'evolvement' should be 'evolution'

Thank you, now we have corrected according with reviewer’s suggestions

Intro para 4 '...a radical change to this views findings consistently indicated..' (this sentence needs to be rewritten)

Thank you, now we have corrected according with reviewer’s suggestions

Intro para 4 should 'autistic individuals' read 'individuals with ASD' (switch in terminology in confusing)

Thank you, now we have corrected according with reviewer’s suggestions

Intro para 7 'similarly in previous study have been described ..' should be '..similarly in a previous study 4 cases with deletions have been described' (please check carefully)

Thank you, now we have corrected according with reviewer’s suggestions

Intro para 10 please remove the word 'kid' and change to 'child' (do this throughout the paper)

Thank you, now we have corrected according with reviewer’s suggestions

Section 1.1: Change to: 'An ectopic pregnancy occurred between the older brother and the child discussed here'

Thank you, now we have corrected according with reviewer’s suggestions

Discussion second sentence: please provide a reference.

Now we added the missing reference.

Discussion (again change 'kid' to 'child')

Thank you, now we have corrected according with reviewer’s suggestions

Discussion para 6 Please provide Manolakos reference

Now we added the missing reference.

Discussion para 7 Please check English and also second last paragraph in Discussion

Now we asked to an expert English speaker to proofread the draft